# Truncated Milk Fat Globule-EGF-like Factor 8 Ameliorates Liver Fibrosis via Inhibition of Integrin-TGFβ Receptor Interaction

**DOI:** 10.3390/biomedicines9111529

**Published:** 2021-10-24

**Authors:** Geun Ho An, Jaehun Lee, Xiong Jin, Jinwoo Chung, Joon-Chul Kim, Jung-Hyuck Park, Minkyung Kim, Choongseong Han, Jong-Hoon Kim, Dong-Hun Woo

**Affiliations:** 1Department of New Drug Development, NEXEL Co., Ltd., 8th Floor, 55 Magokdong-ro, Gangseo-gu, Seoul 07802, Korea; appleagh@nexel.co.kr (G.H.A.); leejh@nexel.co.kr (J.L.); chungjw@nexel.co.kr (J.C.); jckim@nexel.co.kr (J.-C.K.); jhpark@nexel.co.kr (J.-H.P.); sweva@nexel.co.kr (M.K.); nexelceo@nexel.co.kr (C.H.); 2Laboratory of Stem Cells and Tissue Regeneration, Department of Biotechnology, College of Life Sciences and Biotechnology, Science Campus, Korea University, 145 Anam-ro, Seongbuk-gu, Seoul 02841, Korea; 3School of Pharmacy, Henan University, Jin Ming Ave, Kaifeng 475004, China; pierce0701@gmail.com

**Keywords:** liver fibrosis, liver disease, protein therapy, integrin, TGF-β

## Abstract

Milk fat globule-EGF factor 8 (MFG-E8) protein is known as an immunomodulator in various diseases, and we previously demonstrated the anti-fibrotic role of MFG-E8 in liver disease. Here, we present a truncated form of MFG-E8 that provides an advanced therapeutic benefit in treating liver fibrosis. The enhanced therapeutic potential of the modified MFG-E8 was demonstrated in various liver fibrosis animal models, and the efficacy was further confirmed in human hepatic stellate cells and a liver spheroid model. In the subsequent analysis, we found that the modified MFG-E8 more efficiently suppressed transforming growth factor β (TGF-β) signaling than the original form of MFG-E8, and it deactivated the proliferation of hepatic stellate cells in the liver disease environment through interfering with the interactions between integrins (αvβ3 & αvβ5) and TGF-βRI. Furthermore, the protein preferentially delivered in the liver after administration, and the safety profiles of the protein were demonstrated in male and female rat models. Therefore, in conclusion, this modified MFG-E8 provides a promising new therapeutic strategy for treating fibrotic diseases.

## 1. Introduction

Milk fat globule-EGF factor 8 (MFG-E8), also known as lactadherin, is a 46 kDa soluble glycoprotein that plays various roles in physiological and pathological processes, such as facilitating angiogenesis [1], clearing apoptotic cells [2,3], and modulating inflammation [4]. We previously identified a protective role of MFG-E8 in a liver fibrosis model by demonstrating the anti-fibrotic effect of MFG-E8 secreted from mesenchymal stem cells (MSCs) [5]. In addition, it has been reported that MFG-E8 resolves fibrosis by direct binding to collagen and facilitates collagen uptake by macrophages in a bleomycin-induced animal model of idiopathic pulmonary fibrosis [6,7].

Liver fibrosis is a common final pathological process of most chronic inflammatory liver diseases, including nonalcoholic steatohepatitis (NASH) [8,9]. Liver fibrosis has the potential to develop into liver cirrhosis and cancer [10] with mortality rates higher than those of other major cancers (such as lung, colorectal, stomach, and breast cancer) [11]. Therefore, the development of effective and efficient cures for treating liver fibrosis is crucial to reducing mortality. However, despite many trials being conducted over the decades to identify therapeutic molecules, liver fibrosis remains a major cause of death with few therapeutic strategies [12]. In the progression of liver fibrosis, damaged and dead hepatocytes from liver injuries recruit Kupffer cells at their lesion sites; these Kupffer cells secrete substantial quantities of cytokines, including transforming growth factor β1 (TGF-β1), to control liver inflammation [13]. However, the elevated TGF-β1 causes the activation of quiescent hepatic stellate cells (HSCs) that proliferate and become extracellular matrix (ECM)-producing myofibroblast-like cells [14,15]. The activated HSCs further accumulate excessive collagen-rich ECM in the liver, leading to contortions in the normal liver architecture [11]. Therefore, the deactivation or elimination of active HSCs is one of the key factors in resolving liver fibrosis [16].

Integrins are a type of receptor that mediate the interaction between cells and the surrounding ECM, comprising a group of transmembrane cell protein receptors composed of uncooperatively connected α-subunits and β-subunits. These can form at least 24 different combinations that are expressed differently by different cell types and recognize multiple ligands. The expression of integrin in various cell types associated with the development of liver fibrosis and the ability to crosstalk with growth factors and other signal molecules make the concept of targeting integrin an attractive approach to anti-fibrotic treatment [17].

In the present study, we propose a new protein, namely, NP-011, which is a human recombinant protein produced by the structural truncation of MFG-E8. With enhanced therapeutic efficacy and manufacturability, NP-011 could be a promising therapeutic against liver fibrosis.

## 2. Materials and Methods

### 2.1. Production of NP-011 and Confirmation of the Synthesized Protein

The truncation of the Milk fat globule-EGF factor 8 (MFG-E8) structure was designed by NEXEL (Seoul, Korea), and the production was performed by Lugen Sci (Lugen Sci, Bucheon, Gyeonggi-do, Korea). The size of the synthesized NP-011 was confirmed by Coomassie blue staining and Western blot analysis using anti-MFG-E8 antibody (R&D systems, Minneapolis, MN, USA). The purity of NP-011 was analyzed by reversed-phase HPLC using the UltiMate™ 3000 system (Dionex/Thermo Fisher Scientific, Sunnyvale, CA, USA). Briefly, the procedure consisted of two separate steps, with NP-011 diluted in 0.1% (*v*/*v*) formic acid in HPLC-grade water (J.T. Baker^®^) in mobile phase ‘A’ and 0.1% (*v*/*v*) formic acid in HPLC-grade acetonitrile in mobile phase ‘B’, consisting of 0.1% (*v*/*v*) formic acid in HPLC-grade acetonitrile (Honeywell Burdick & Jackson). The flow rate was set to 100 μL/min during analysis, and the wavelength of protein peaks was obtained at 280 nm. The entire procedure was performed at 60 °C ± 1 °C using a ZORBAX 300SB-C8 (2.1 mm × 50 mm, 3.5 μm) column (Agilent, Santa Clara, CA, USA). The data were analyzed using Chromeleon™ software (Ver. 6.8 SR 10, Thermo Fisher Scientific).

### 2.2. Rodent Fibrosis Model Induction and Efficacy Tests of NP-011 

To compare the efficacy of Milk fat globule-EGF factor 8 (MFG-E8) and NP-011, 200 mg/kg of TAA (thioacetamide, Sigma-Aldrich, St. Louis, MO, USA) was administrated into 5- to 6-week-old male C57BL/6 mice (*n* = 4 for each group, three times per week for 8 weeks) and 160 μg/kg of each protein was administrated intraperitoneally. The mice were sacrificed at 3 days after the administration of proteins for analysis. Various doses of NP-011 (20–160 μg/kg, *n* = 5 for each group) were further tested in the same TAA-induced liver fibrosis model. To test the anti-fibrotic effects of multiple administrations of NP-011, TAA was injected into the mice for 12 weeks, and then 40 μg/kg of NP-011 was intraperitoneally administered into the TAA-induced liver fibrosis model one to six times at 5-day intervals (*n* = 4 for each group). For the progressive liver fibrosis model, TAA was injected into the mice for 4 weeks, then 40 μg/kg of NP-011 and TAA were intraperitoneally co-administered into the mice three times a week for another 4 weeks (*n* = 5 for each group). 

### 2.3. Histological Analysis and Immunofluorescence Assay 

The liver tissues were fixed in 4% paraformaldehyde (PFA) and dehydrated in a graded ethanol series. The tissues were then cleared in xylene and embedded in paraffin. Paraffin-embedded tissue sections were stained with hematoxylin and eosin (H&E, Abcam, Cambridge, MA, UK) and Sirius red (American MasterTech Scientific, Lodi, CA, USA) for the evaluation of liver fibrosis. For immunofluorescence staining, sectioned tissues underwent antigen retrieval with citric acid, and tissues blocked with 10% donkey serum containing PBS were probed with the primary antibody against alpha-smooth muscle actin (α-SMA) or albumin (ALB) at 4 °C overnight. For visualization of the staining, the sections were washed with 0.1% bovine serum albumin (BSA) containing phosphate-buffered saline (PBS) and stained with fluorescently labeled secondary antibodies (Invitrogen/Thermo Fisher Scientific, Carlsbad, CA, USA). Digital images were captured using a microscope (Nikon corporation, Tokyo, Japan) and analyzed using ImageJ software. 

### 2.4. Quantitative Reverse-Transcription PCR (RT-qPCR)

Total RNA was extracted using TRIzol (Invitrogen), and cDNA was synthesized from 1 μg of total RNA using a ReverAidTM H Minus First Strand cDNA Synthesis Kit (Invitrogen) according to the manufacturer’s protocol. Subsequent polymerase chain reaction was carried out using AccuPower^®^ PCR-Premix (Bioneer, Daejeon, Korea) and the DNA Engine Peltier Thermal Cycler (Bio-rad). Quantitative polymerase chain reaction (qPCR) was performed using the CFX96 real-time PCR detection system (Bio-Rad, Hercules, CA, USA) with iQ™ SYBR^®^ Green Supermix (Bio-Rad). The specific primers used are provided in Appendix A. The mRNA levels were normalized to the level of GAPDH (glyceraldehyde-3-phosphate dehydrogenase).

### 2.5. Whole Transcriptome Analysis of Mouse Livers 

Six-week-old male C57BL/6N mice were purchased from Koreabio (DBL, Seoul, Korea). Experimental protocols concerning the use of laboratory animals were reviewed by the Korea University Institutional Animal Care & Use Committee (KUIACUC) and approved (KUIACUC-2018-78). Animals were fed a standard diet with free access to water. In the sham group (*n* = 18), mice received 200 mg/kg body weight of thioacetamide (TAA, Sigma, St. Louis, MO, USA) by intraperitoneal (I.P.) injection for 8 weeks. In the control group (*n* = 3), mice received the same volume of normal saline. In the NP-011-administered group, protein (NP-011, *n* = 3, 160 μg/kg body weight) was administered by I.P. injection on the last day of TAA injection. Mice were sacrificed by CO_2_ inhalation after 24 h. The liver was removed immediately for RNA-seq. The whole liver was homogenized in cold TRIzol (Sigma, St. Louis, MO, USA) and stored below −80 °C. For RNA seq analysis, these samples were sent to BGI Tech Solutions Company (BGI Tech, Shenzhen, Guangdong, China). GSEA analysis was conducted using GSEAv17 (Broad Institute, Cambridge, MA, USA).

### 2.6. Cell Culture and In Vitro Fibrosis Modeling

The human hepatic stellate cell (HSC) line, hTERT-HSC, was cultured in Dulbecco’s modified Eagle’s medium (DMEM; GE Healthcare Life Sciences, Marlborough, MA, USA) supplemented with 10% fetal bovine serum (FBS; Gibco/Thermo Fisher Scientific, Waltham, MA, USA), 100 U/mL penicillin, and 100 mg/mL streptomycin (Gibco). The human HEK-293FT cells kindly provided by Prof. Hyunggee Kim (Korea University, Seoul, Korea) were cultured in Dulbecco’s modified Eagle’s medium (DMEM; GE Healthcare Life Sciences, Chicago, IL, USA) supplemented with 10% fetal bovine serum (FBS; Gibco, New York, NY, USA), 100 U/mL of penicillin, and 100 mg/mL of streptomycin (Gibco). To investigate the effects of NP-011 on TGF-β1-mediated HSC activation, human HSC lines (hTERT-HSCs) were grown in the presence of serum, then starved in DMEM containing 0.2% FBS for 24 h before TGF-β1 treatment. The serum-starved HSCs were pretreated with 10 ng/ml TGF-β1 for 1 h, and the HSCs were exposed to 100–1500 ng/mL NP-011 for 6 h. To block integrin αvβ3 and αvβ5 in the HSCs, the HSCs were pretreated with 1 μM Cilengitide (Selleck Chemicals, Houston, TX, USA) for 2 h before treatment with TGF-β1. The activation and deactivation of HSCs was quantitatively determined by 5-ethynyl-2′-deoxyuridine (EdU) assay. In order to establish a human liver fibrosis model, the liver spheroids were formed by a mixture of hepatocytes (Hepatosight-S^®^, NEXEL, Seoul, Korea) and hTert-HSCs used in our previous research [5] in an ultra-low-attachment 96-well plate (Corning) with a cell density ratio of 2 to 1, respectively. The liver spheroids were cultured for 21 days, and 50 mM of acetaminophen (APAP) was applied to induce fibrosis. To test the efficacy of NP-011 against the APAP-induced 3D liver fibrosis model, 500 ng/ml of NP-011 was applied for 48 h. 

### 2.7. TGF-β Luciferase Signaling Reporter Assay 

The luciferase signaling reporter assay was performed according to the manufacturer’s protocol (Qiagen, Hilden, Germany). We selected the cell line HEK-293FT, and Attractene Transfection Reagent (Qiagen) was used for transfection. After 18–20 h of transfection, we changed the medium to a complete growth medium (DMEM with 10% FBS, 0.1 mM NEAA, 1 mM sodium pyruvate, 100 U/mL penicillin, and 100 μg/mL streptomycin) for 30 min of incubation. Then, the cell medium was changed to assay medium (Opti-MEM^®^ containing 0.5% fetal bovine serum, 1% NEAA, 100 U/mL penicillin, and 100 μg/mL streptomycin) including 500 ng/mL of NP-011 and Milk fat globule-EGF factor 8 (MFG-E8). After that, the luciferase assay was carried out using the Dual-Glo Luciferase Assay System (Promega). To measure firefly luciferase activity, we added a volume of Dual-Glo^®^ Luciferase Reagent equal to the culture medium volume to each well and mixed. After at least 60 min, the firefly luminescence was measured. After that, a volume of Dual-Glo^®^ Stop & Glo^®^ Reagent equal to the original culture medium volume was added to each well. After at least 10 min, the luminescence was measured. We calculated the ratio of luminescence from the experimental reporter to luminescence from the control reporter. We then normalized this ratio to the ratio of a control well or series of control wells that were treated consistently on all plates.

### 2.8. Western Blot Analysis and Immunoprecipitation (IP) Analysis 

Protein samples were prepared by solubilizing HSCs in RIPA lysis buffer (LPS solution) containing proteinase inhibitors (Roche, Basel, Switzerland). A total of 40 μg of protein from cells was separated by SDS-PAGE (Bio-Rad) and transferred to PVDF transfer membranes (Pall Corporation, Port Washington, NY, USA). The membranes were incubated for 60 min with 5% skim milk in TBS-T (10 mM Tris-HCl pH 7.9, 150 mM NaCl, and 0.05% Tween-20) to block nonspecific antibody binding sites. After blocking, the membranes were immunoblotted with primary antibodies overnight at 4 °C. The antibodies used in the present study are provided in Appendix A. To detect each band in Western blot, the membranes were incubated for 2 h with horseradish peroxidase (HRP)-conjugated secondary antibodies (Thermo Fisher Scientific, Waltham, MA, USA) at room temperature. After rinsing with TBS-T, the membranes were developed with the Pierce™ ECL Western blotting substrate (Thermo Fisher Scientific, Waltham, MA, USA) and bands were detected using a chemiluminescence imaging system (GE Healthcare Life Sciences, Chicago, IL, USA). For immunoprecipitation (IP) analysis, a total of 400 μg of protein was incubated at 4 °C for 12 h with 1 μg of TGFβRI antibody, conjugated to protein A/G sepharose beads (Santa Cruz Biotechnology, Inc., Dallas, TX, USA) washed in lysis buffer, then separated on SDS-PAGE gels. To detect each band in Western blot and IP analysis, the membranes were incubated for 2 h with HRP-conjugated secondary antibodies (Thermo Fisher Scientific, Waltham, MA, USA) at room temperature. After rinsing with TBS-T, the membranes were developed with the Pierce™ ECL Western blotting substrate (Thermo Fisher Scientific, Waltham, MA, USA) and bands were detected using a chemiluminescence imaging system (GE Healthcare Life Sciences, Chicago, IL, USA). 

### 2.9. EdU Incorporation Assays 

For the EdU incorporation assay, human HSCs were seeded at 2 × 10^4^ cells per well in 12-well plates and cultured for 24 h. After treatment with TGF-β1 and/or NP-011, the serum-starved HSCs were incubated with EdU (10 μM) for an additional 6 h, and EdU incorporation was accessed using the Click-iT EdU Imaging Kit (Thermo Fisher Scientific, Waltham, MA, USA) according to the manufacturer’s instructions. Digital images of EdU-positive cells were captured using a microscope (Nikon) and analyzed using ImageJ software.

### 2.10. Proximity Ligation Assay (PLA) 

For the PLA, human HSCs were seeded at 2 × 10^4^ cells per well on 18 mm circular cover glass in 12-well plates and cultured for 24 h. After serum starvation for 24 h, the cells were treated with 10 ng/mL of TGF-β1 and/or 500 ng/mL of NP-011, then further incubated for 30 min. The PLA incorporation was accessed using the Duolink In Situ Red Starter Kit (Merck, Kenilworth, NJ, USA) according to the manufacturer’s instructions. Digital images of PLA-positive cells were captured using a microscope (Nikon) and analyzed using ImageJ software (https://imagej.nih.gov/ij/, accessed on 29 June 2020).

### 2.11. Radioligand Binding Assay

To analyze the binding affinity between NP-011 and integrins, we performed a radioligand binding assay, and an assay was done by Gifford Bioscience (Birmingham, UK). Briefly, the NP-011 was radio-labeled with iodine-125 (^125^I). Then, ^125^I-labeled NP-011 in buffer (50 mM Tris, 5 mM MgCl_2_, 100 mM NaCl, 1% BSA (pH 7.4)) was incubated in a HIS-tagged αvβ3 (ACRO Biosystems IT3-H52E3) or αvβ5 (ACRO Biosystems IT5-H52W5) protein-coated plate with a concentration range of 0–100 μg/mL. The incubation was stopped by washing the wells with wash buffer (50 mM Tris, 5 mM MgCl2, pH 7.4, ice cold). Following washing, NaOH (0.1 M) was added to each well and the plates were incubated at 40 °C for 1 h to digest the protein. Following digestion, the samples were transferred to a counting plate and neutralized, then scintillation cocktail (Betaplate Scint; PerkinElmer, Waltham, MA, USA) was added and the radioactivity counted in a Wallac^®^ TriLux 1450 MicroBeta counter. All experiments were validated step by step by Gifford Bioscience. Data analysis was performed using the nonlinear curve fitting routines in Prism^®^ (GraphPad Software Inc, GraphPad Prism 5.0, San Diego, CA, USA) to obtain K_d_ values.

### 2.12. Collagenase Activity Assay 

The collagenase activity assay was performed according to the manufacturer’s instruction (Chondrex, Woodinville, WA, USA). Briefly, enzyme components were mixed with the reaction solution, and the enzymatic reaction was initiated by mixing with 1.0 mg/mL FITC-labeled bovine collagen I substrate, followed by incubation at 37 °C for 1 h. The enzymatic reactions without the collagen substrate or the enzyme components were used as a negative control. To stop the enzymatic reaction, 10 mM o-phenanthroline and 38.5 μM elastase were added to samples, followed by incubation at 37 °C for 10 min. Finally, the extraction buffer was mixed with the reaction solutions. The supernatant was used for the measurement with the spectrofluorometer.

### 2.13. Statistical Analysis 

To evaluate the anti-fibrotic effects of NP-011 in vivo, at least three animals per group were used in each experiment, and data were obtained from two or three independent experiments. The percentages of positive areas for the Sirius red staining or immunostaining of the total image area were measured using ImageJ software and expressed as relative values compared to those in normal livers or control cell cultures. Student’s *t*-test was used to analyze the statistical significance of differences between the paired groups. One-way analysis of variance (ANOVA) was used to test the statistical significance of differences among multiple groups (more than two groups). The data are expressed as the means ± SEM of at least three independent experiments, and all statistical tests were two-sided; data with *p* < 0.05 or *p* < 0.01 were assumed to be statistically significant.

## 3. Results

### 3.1. Structural Truncation of MFG-E8 Enhances the Anti-Fibrotic Effect of MFG-E8

The structure of human Milk fat globule-EGF factor 8 (MFG-E8) includes three domains: the signaling peptides of N-terminals, the epidermal growth factors (EGF) with an arginine–glycine–aspartic acid (RGD) motif, and the C domains (C1 and C2) [2,3]. Although it is well known that MFG-E8 regulates inflammatory responses by RGD motif binding to immune cells and engulfing phosphatidylserine (PS)-expressing apoptotic cells, it is unclear how MFG-E8 is responsible for the anti-fibrotic effect [18,19]. A recent report showed that the glycosylation-bearing C2 domain of MFG-E8 plays a key role in recognizing PS in apoptotic cells [20]. Therefore, we hypothesized that the removal of the C2 domain in MFG-E8 might enhance the anti-fibrotic power of the MFG-E8 protein, and we therefore synthesized NP-011 (EGF + C1 domain), a truncated form of MFG-E8 (Figure 1A). Coomassie brilliant blue (CBB) staining and Western blot analysis confirmed the ~25 kDa size of the synthesized NP-011 from two separate productions, and the result of reverse-phase HPLC showed 80.7% and 97.2% purity for the two batches (Figure 1B). In the following experiments, Batch #2 with over 95% purity was used.

A subsequent efficacy test in a thioacetamide (TAA)-induced liver fibrosis mouse model (Figure 1C) revealed that the administration of commercially available MFG-E8 effectively reduced the fibrotic area (Sirius red-stained area, Figure 1D,E) and downregulated the expression level of liver-fibrosis-related genes (Col1a1, Col1a2) (Figure 1F), as we previously demonstrated [5]. However, it is compelling that the administration of NP-011 eliminated the fibrotic area more substantially and downregulated the expression level of fibrotic genes in the injured liver more significantly than did the administration of MFG-E8 (Figure 1D,E). These results indicate that MFG-E8 with deletion of the C2 domain gives more beneficial effects in curing liver fibrosis than does the original protein.

### 3.2. NP-011 Significantly Reverses Liver Fibrosis at Minimal Dosage 

To explore the effective dosage of NP-011, different doses of NP-011 (20 μg/kg, 40 μg/kg, 80 μg/kg, and 160 μg/kg) were administered in a TAA-induced liver fibrosis model, and the fibrotic factors were analyzed 3 days after NP-011 administration (Figure 2A). The administration of TAA significantly increased the fibrosis area in the mouse liver (Figure 2B,C, Sham). In contrast to sham-treated liver tissues, all the NP-011-administrated groups, ranging from 20 to 160 μg/kg, showed remarkably diminished fibrosis area (Figure 2B,C). Consistent with the decrease in fibrosis, a key fibrosis marker, Acta2 (α-SMA, a marker for myofibroblast-like cells differentiated from HSCs), and other fibrosis-related genes were markedly and significantly downregulated after NP-011 administration (Figure 2D,E). It was noteworthy that the expression of integrin families (integrin αv, integrin β3, and integrin β5) increased as the fibrosis progressed but decreased after NP-011 treatment in the liver of the TAA-induced model (Figure 2D). Taken together, these results imply that NP-011 shows great therapeutic efficacy at low dosage ranges; most effective was the administration of 40 μg/kg of NP-011, resulting in a constant reduction of α-SMA expression in the injured liver.

### 3.3. The Efficacious Dose of NP-011 Shows Therapeutic Efficacy in Different Models Associated with Fibrosis 

The minimum efficacious dose of NP-011 (40 μg/kg) was further tested in different liver fibrosis models. Firstly, the efficacy of repeated administrations of NP-011 was tested in a chronic model of liver fibrosis. For this, TAA injections were extended from 8 to 12 weeks (three times a week), and 40 μg/kg of NP-011 was then administered to the mice one to six times at 5-day intervals (Figure 3A). The 12-week TAA injections resulted in the development of sustained fibrosis areas in the liver despite the TAA injections being halted 30 days before the mice were sacrificed (Figure 3B, sham). In contrast, the administration of NP-011 significantly resolved the fibrotic areas in injured livers, and the fibrotic regression was positively correlated with the number of times of administration (Figure 3B). 

Liver fibrosis is a progressive disease; thus, evaluation of the efficacy of NP-011 on progressing liver fibrosis may provide a better translation of the animal study to the clinical realm. To confirm the therapeutic efficacy of NP-011 in progressing liver fibrosis, NP-011 (40 μg/kg, three times a week) was concurrently administrated with TAA for the last 4 weeks of model induction (Figure 3C). As expected, the concurrent administration of TAA and NP-011 markedly diminished the fibrosis areas in the livers, compared with those in the livers that received TAA only (Figure 3D). We also used an acetaminophen (APAP)-induced in vitro human fibrosis model and found that NP-011 significantly reduced APAP-induced HSC activation in 3D hepatic spheroids consisting of hepatocytes and hepatic stellate cells (Figure 3E). Therefore, NP-011 shows therapeutic efficacy not only in various fibrotic models in animals, but also in a human liver fibrotic model.

The effects of NP-011 on the expression of pro-fibrotic MMP2 and collagenase activity in HSCs were further tested based on the previous finding that showed secretion of pro-fibrotic MMP2 and expression of collagenase mRNA in rat HSCs [21,22]. As reported, the expression of MMP2 mRNA in HSCs was increased after TGF-β1 treatment. However, NP-011 treatment of TGF-β1-treated HSCs significantly down-regulated the increased expression of MMP2 (Appendix A). Interestingly, collagenase activity was increased in HSCs after NP-011 treatment in the presence or even absence of TGF-β1, while there was no effect with TGF-β1 treatment only (Appendix A), indicating that NP-011 itself has a capacity to increase collagenase activity in deactivated HSCs.

### 3.4. NP-011 Deactivates HSCs through the Suppression of TGF-β/Smad 2 Signaling and Prevents Fibrogenesis in Human Hepatic Stellate Cells via the Inhibition of Integrin–TGFβ Receptor Interaction

We next investigated the action mechanism of NP-011 underlying the resolution of fibrosis in the liver. As is well known, TGF-β contributes to liver fibrosis by the induction of epithelial-to-mesenchymal transition (EMT) in hepatocytes [23] and activation of HSCs to myofibroblasts [24]. Thus, effective suppression of TGF-β in liver disease environments is a key factor for treating liver fibrosis. Signaling reporter assay revealed that NP-011 rapidly suppressed TGF-β signaling within 30 min, whereas Milk fat globule-EGF factor 8 (MFG-E8) could suppress TGF-β signaling 2 h after treatment (Figure 4A), implying the enhanced capacity of NP-011 for inhibiting TGF-β signaling in liver disease. Western blot analysis further confirmed the attenuation of Smad 2 phosphorylation, a downstream molecule of TGF-β signaling, in TGF-β1-treated HSCs (Figure 4B).

Previous reports demonstrated that integrin αvβ3 and αvβ5 on the surface of HSCs are key factors in regulating fibrosis [5,25], and crosstalk between integrins and TGF-β signaling in the regulation of pathological EMT and myofibroblast differentiation has also been reported [26]. Furthermore, in this study, integrin αv, integrin β3, and integrin β5 in the liver of TAA-induced liver fibrosis mice were increased, but they were decreased again in the liver of the NP-011 administered liver fibrosis model (Figure 2D). Thus, targeting TGF-β related integrins could be another contributing factor in treating liver fibrosis. As shown in Figure 4C, NP-011 treatment in TGF-β1-treated HSCs suppressed the proliferation of HSCs and returned the proliferative HSCs to quiescent status. However, treatment with integrin αvβ3/αvβ5 inhibitor in the presence of NP-011 inhibited the suppressive role of NP-011 in the proliferation of TGF-β1-treated HSCs (Figure 4C). These results suggest that NP-011 directly binds to integrin αvβ3 and αvβ5 and interferes with integrin αvβ3 and αvβ5 interactions with TGF-β signaling.

Proximity ligation assay (PLA) revealed the direct physical associations between TGF-βRI and integrin β3 and β5, and the interactions became stronger upon TGF-β1 treatment in HSCs (Figure 4D,E). However, NP-011 treatment in TGF-β1-treated HSCs significantly loosened the associations between TGF- βRI and integrin β3 and β5 (Figure 4D,E). These physical associations were further confirmed by immunoprecipitation assay, and these patterns were identical to the Smad2 phosphorylation pattern by TGF and/or NP-011 treatment (Figure 4B). Finally, we determined the direct binding of NP-011 to integrin αvβ3 and αvβ5 by radioligand binding assays and confirmed the NP-011 binding to integrin β3 and β5 with calculated Kd values of 50.4 nM and 2.0 nM, respectively. Notably, NP-011 showed about 12-fold stronger binding to integrin β5 than did MFG-E8 (Kd of 25.4 nM, Appendix A). NP-011 did not show binding affinity to other fibrosis-related integrin families such as integrin β1 and β6 (data not shown). These results suggest that NP-011 specifically binds to integrin β3 and β5 and interferes in the interaction between integrins and TGF-βRI, resulting in suppression of the TGF-β cascade and a decrease in HSC proliferation.

### 3.5. Bio-Distribution and Safety Profiles of NP-011 

The bio-distribution and safety of NP-011 were further assessed in a rodent model. When NP-011 was intravenously administrated into mice, the administrated NP-011 was preferentially delivered into the liver; about 48% and 58% of the administrated NP-011 were detected in the liver within 30 min and 60 min, respectively (Appendix A). Furthermore, no adverse effects were observed in hematology and biochemistry analysis of blood serum in male and female rats when 0.2 mg/kg or 2 mg/kg of NP-011 was intravenously administrated into rats daily for 4 weeks (Appendix A). No measurable changes in hepatic or renal functions have been observed after the use of Milk fat globule-EGF factor 8 (MFG-E8) in many animal disease models [27,28,29,30,31,32], so the truncation of the C2 domain in MFG-E8 did not alter the safety profiles in vivo. Thus, NP-011 preferentially targets the liver after its administration and has excellent safety profiles.

## 4. Discussion

Our previous study demonstrated a reduction in mouse liver fibrosis with the administration of recombinant MFGE8 [5]. Therefore, this study aimed to increase the improvement effect on liver fibrosis by utilizing a structural change in Milk fat globule-EGF factor 8 (MFG-E8) and to identify a clear mechanism. 

The C2 domain of MFG-E8 facilitates binding to phosphatidylserine (PS) on apoptotic cells and integrin αvβ3/αvβ5 on phagocytic cells as a bridging molecule [4,33,34]. By truncation of the C2 domain in MFG-E8, NP-011 might be considered to have weaker binding to PS, but it might have strengthened binding affinity with integrins through the RGD motif as a compensatory mechanism. In fact, the binding affinity of NP-011 to integrin αvβ5 was significantly increased (>10-fold) compared to the binding of MFG-E8 (Figure 4F and Appendix A). It was reported that the inhibition of integrin αvβ5 reduced the activation of TGF-β signaling by 66%, twice as much as that by blocking αvβ3 or αvβ1 integrin [35]. Therefore, the removal of the C2 domain in MFG-E8 protein might compensate for stronger binding to integrin αvβ5, thereby inhibiting TGF-beta signaling more effectively. 

MFG-E8 protein has therapeutic effects in many diseases but has also some limitations in development. For example, MFG-E8 contains a medin site [36], known to cause Alzheimer’s [37], type 2 diabetes [38], and aging [39], and glycosylation sites that make it difficult to synthesize with potential immunogenicity after administration in the body; both of these are present in the C2 region. Thus, through the truncation of the C2 domain in MFG-E8, NP-011 might be free from the above concerns in clinical applications. With these advantages, the anti-fibrotic power of NP-011 was previously demonstrated in an idiopathic pulmonary fibrosis (IPF) model, and it showed efficacy comparable to that of existing drugs (nintedanib and pirfenidone) for treating IPF patients [7].

In the present study, our results demonstrated therapeutic effectiveness in liver fibrosis mouse models induced by various methods. This shows that NP-011 improves fibrosis, no matter the source of liver fibrosis. It was also impressive that TGF-β/Smad2 signaling was reduced, but no toxicity was found, and the association with integrin was confirmed. This is a very important result in the absence of drugs that restore fibrosis to its original state. Recently, research on integrins in fibrotic diseases has been actively conducted. In particular, the association of fibrosis with RGD-binding integrin has been reported [40,41], and the importance of targeting integrin αv in tissue fibrosis was also reported [42]. As shown in the results of this study, NP-011 was preferentially delivered in the liver and specifically bound to integrin αvβ3 and αvβ5 with enhanced binding affinity compared to that of MFG-E8 (data not shown). Furthermore, NP-011 interfered with the interactions between integrins and TGF-βRI on HSCs in the presence of TGF-β1, and it suppressed TGF-β/Smad2 signaling in HSC culture and deactivated the proliferation of HSCs activated by TGF-β1 treatment in the culture. Since NP-011 bound to the integrins more strongly than MFG-E8, it is also demonstrated that NP-011 more efficiently suppressed TGF-β signaling than did MFG-E8 in this study. Because HSCs have pivotal roles in fibrogenesis and further progression of fibrosis in the liver [14,15], targeting HSCs could be a key factor in resolving liver fibrosis. Therefore, NP-011 has promising potential to target HSCs effectively and to resolve liver fibrosis with beneficial effects over the original form of MFG-E8.

Finally, high manufacturability is another advantage of NP-011 in clinical applications. NP-011 has a smaller protein structure than MFG-E8 and can be mass-produced without difficulty in synthesis. Furthermore, NP-011 is produced as a secretory form (authentic NP-011) without methionine at the N-terminus of the protein with any tag (e.g., FLAG-tag or his-tag) and random glycosylation (data not shown), which could induce significant problems upon administration of medications [43]. Thus, in conclusion, NP-011 could provide a highly effective and reliable new protein therapy for treating liver fibrosis.

## 5. Conclusions

Collectively, the structural truncation of Milk fat globule-EGF factor 8 (MFG-E8) enhanced its therapeutic efficacy against liver fibrosis with benefits of rapid/effective suppression of TGF-β signaling associated with integrin binding for deactivating HSCs and high productivity without concerns in clinical applications. Therefore, the modified MFG-E8 (NP-011) could be an advanced and promising new drug candidate, even in the presence of natural MFG-E8 protein.

## 6. Patents

NP-011 and the use of NP-011 for treating liver fibrosis is protected by published or unpublished patents (KOR/10-1947902, PCT/KR2017/005150, EU/17870624.8, JP/6585296, US/15/994.323, CN/201780004259.5, JP/2019-160324, and PCT/IB2019/001136), and these intellectual property rights belong to NEXEL. Co., Ltd.

## Figures and Tables

**Figure 1 biomedicines-09-01529-f001:**
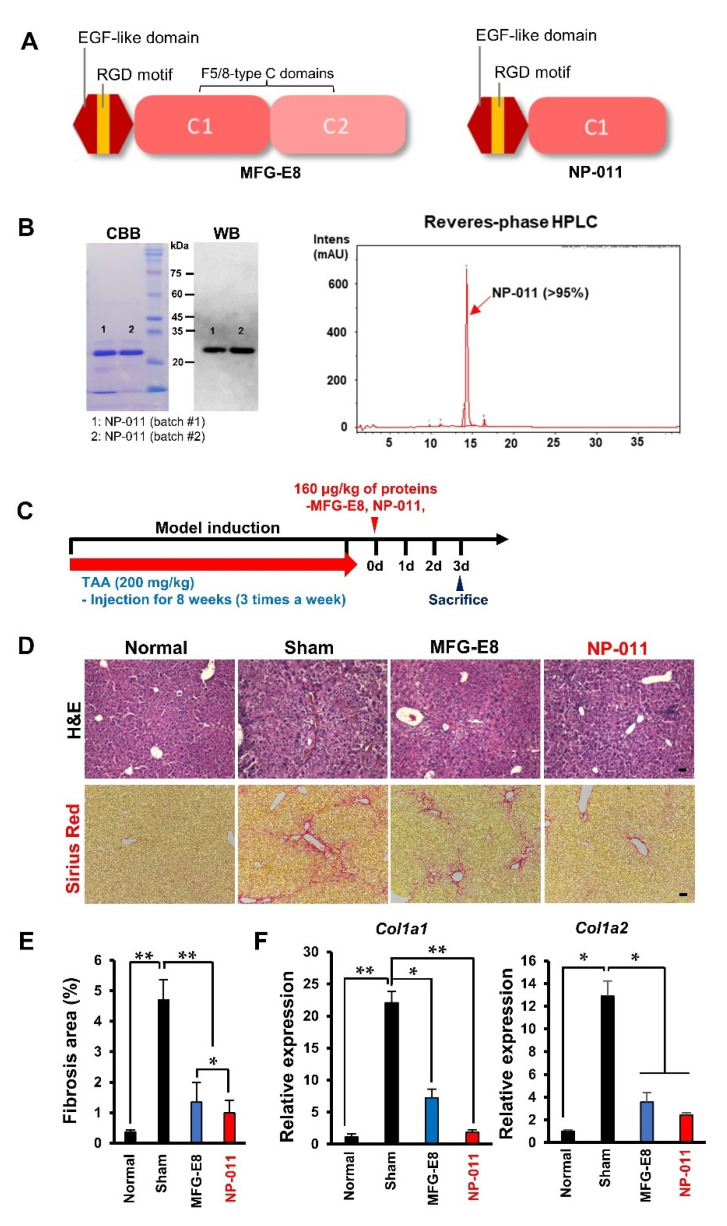
Production of truncated MFG-E8 and the advantages of NP-011 for treating liver disease. (**A**) Structural comparison of MFG-E8 and NP-011. (**B**) Confirmation of synthesized NP-011. Coomassie brilliant blue (CBB) staining and Western blot analysis showed ~25 kDa sized truncated forms of MFG-E8 (NP-011). The purity of the produced NP-011 protein (Batch #2) was analyzed by reverse-phase HPLC analysis. (**C**) Overall schematic schedule for testing the efficacy of the human recombinant proteins in a liver fibrosis model. The red arrow indicates the period for TAA administrations for inducing liver fibrosis. (**D**) Representative images of histological analysis (H&E and Sirius red staining) for the liver of normal mice, a TAA-induced liver fibrosis model (Sham), and a protein (MFG-E8, NP-011) administered liver fibrosis model. Scale bar, 200 μm. (**E**) Comparison of the quantitative fibrotic area in the livers of the mice. (**F**) Comparison of mRNA expression for fibrotic markers (*Col1a1*, *Col1a2*) in the liver of normal mice, a TAA-induced liver fibrosis model (Sham), and a protein (MFG-E8, NP-011) administered liver fibrosis model. Bars represent the means ± SD from three replicates in each group. * *p* < 0.05, ** *p* < 0.01, ANOVA followed by Tukey’s multiple comparison test.

**Figure 2 biomedicines-09-01529-f002:**
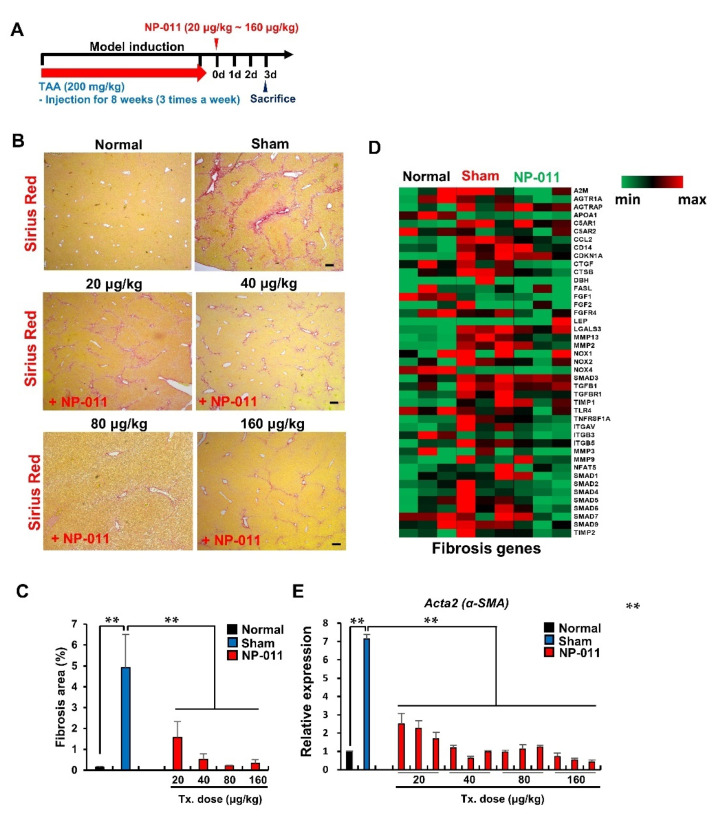
Resolution of fibrosis by administration of various doses of NP-011. (**A**) Overall schematic schedule for testing the efficacy of the NP-011 in a liver fibrosis model. The red arrow indicates the period for TAA administrations for inducing liver fibrosis. (**B**) Representative images for comparing the fibrotic areas (Sirius red stained areas) in the liver of normal mice, a TAA-induced liver fibrosis model, and an NP-011-administered liver fibrosis model at various doses (20–160 μg/kg). Scale bar, 200 μm. (**C**) Quantitative analysis of the fibrotic areas in the livers of tested mice in B. Tx. indicates treatment. Bars represent the means ± SD from four mice in each group. ** *p* < 0.01, Student’s *t*-test. (**D**) Heatmap data show fibrosis-relevant genes observed in livers from normal mice, TAA-induced mice, and NP-011-administered liver fibrosis mice. (**E**) Comparison of α-SMA (*Acta2*) expression in the liver of normal mice, a TAA-induced liver fibrosis model, and an NP-011-administered liver fibrosis model by quantitative PCR. Tx. indicates treatment. ** *p* < 0.01, Student’s *t*-test.

**Figure 3 biomedicines-09-01529-f003:**
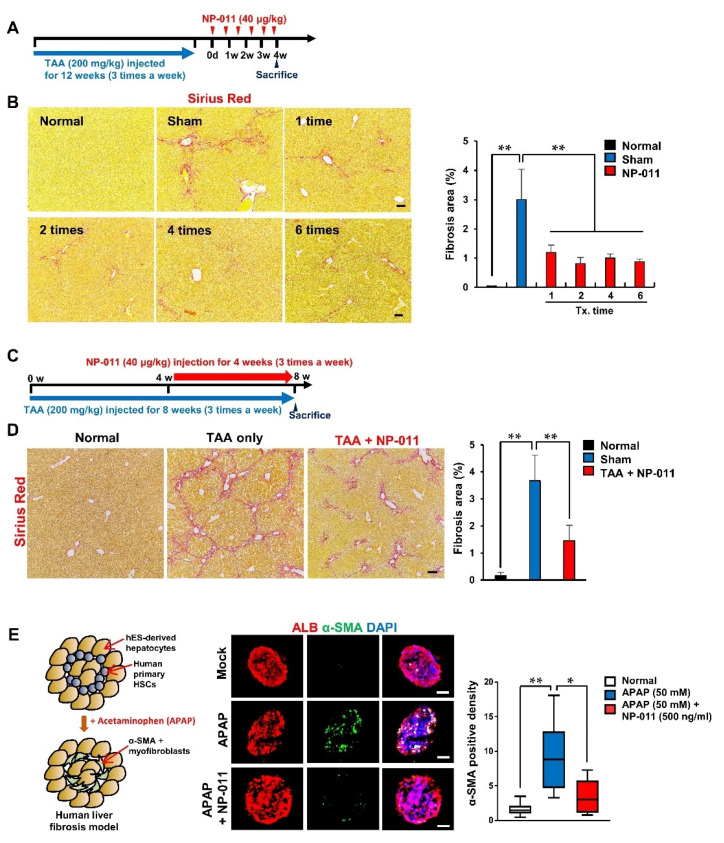
Therapeutic efficacy of a minimal effective dosage of NP-011 in various model inductions. (**A**) Overall schematic schedule for testing the efficacy of NP-011 in an advanced liver fibrosis model. The blue arrow indicates the period for TAA administrations for inducing liver fibrosis. (**B**) Representative images for analyzing fibrotic areas in livers of normal mice, a TAA-induced liver fibrosis model, and a 40 μg/kg NP-011-administered liver fibrosis model with different numbers of administrations (1, 2, 4, and 6 times). Scale bar, 50 μm. Quantitative analysis of fibrotic areas in the livers of normal mice, a TAA-induced liver fibrosis model, and a 40 μg/kg NP-011-administered liver fibrosis model presented in the left panel. Bars represent the means ± SD from four mice in each group. ** *p* < 0.01, Student’s *t*-test. (**C**) Overall schematic schedule for testing the efficacy of NP-011 in a progressing liver fibrosis model. The blue arrow indicates the period for TAA administrations for inducing liver fibrosis. The red arrow indicates the period for NP-011 administrations. (**D**) Representative images for analyzing fibrotic areas in livers of normal mice, a TAA-induced liver fibrosis model, and a TAA-injected liver fibrosis model with co-administration of NP-011. Scale bar, 100 μm. Quantitative analysis of fibrotic areas of livers in normal mice, a TAA-induced liver fibrosis model, and a TAA-injected liver fibrosis model with co-administration of NP-011 presented in (**D**). Bars represent the means ± SD from five mice in each group. ** *p* < 0.01, Student’s *t*-test. (**E**) Left panel: Graphical description representing a human liver fibrosis model using human embryonic stem cell (hESC)-derived hepatocytes and human primary HSCs treated with APAP. Middle panel: Representative images for analyzing hepatocytes and activated myofibroblasts with ALB (red) and α-SMA (a biomarker for fibrosis, green). Scale bar, 50 μm. Right panel: Quantitative analysis of the percentage of α-SMA positive cells in the human liver fibrosis model. Bars represent the means ± SD from five replicates in each group. ** *p* < 0.01, * *p* < 0.05 Student’s *t*-test.

**Figure 4 biomedicines-09-01529-f004:**
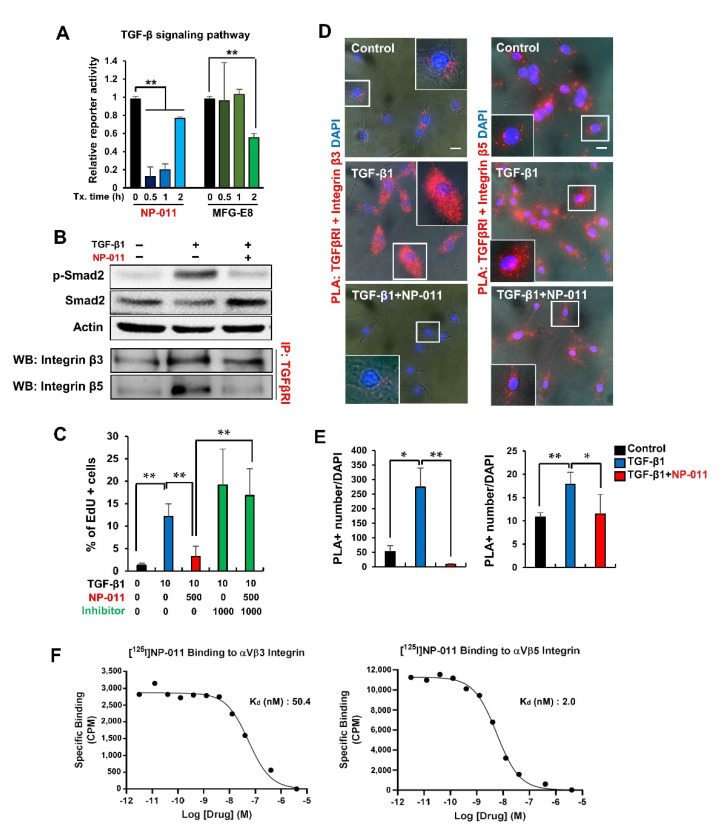
Mechanistic study of NP-011 in the resolution of liver fibrosis. (**A**) Luciferase reporter assay confirming TGF-β signaling pathway transcriptional activity from human HEK-293FT cells after 0.5 h, 1 h, or 2 h of treatment with NP-011 (500 ng/mL) or MFG-E8 (500 ng/mL). Bars represent the means ± SD from three replicates in each group. * *p* < 0.05, ** *p* < 0.01, Student’s *t*-test. (**B**) Western blot (WB) analysis of Smad2 phosphorylation in human HSCs in the presence of TGF-β1 (10 ng/mL) and/or NP-011 (500 ng/mL), and immunoprecipitation (IP) analysis of physical associations between TGF-βRI and integrin β3/β5 in the presence of TGF-β1 (10 ng/mL) and/or NP-011 (500 ng/mL). (**C**) EdU incorporation assay for cell proliferation of human HSCs in the presence of TGF-β1, NP-011, and/or Cilengitide trifluoroacetate (CT, inhibitor of integrin ανβ3 and ανβ5). Bars represent the means ± SD from five replicates in each group. ** *p* < 0.01, Student’s *t*-test. (**D**,**E**) PLA assay for studying the physical interaction between TGFBRI and integrin αvβ3/αvβ5 after TGF-β1 (10 ng/mL) treatment with/without NP-011 (500 ng/mL). Red signals indicate the interactions between TGFBRI and integrin β3 (left) and β5 (right). Quantitative analysis of the number of red signals in each cell. Bars represent the means ± SD from five replicates in each group. * *p* < 0.05, ** *p* < 0.01, Student’s *t*-test. Scale bar, 20 μm. (**F**) Radioligand binding assay to determine the binding affinity of NP-011 to immobilized integrin αVβ3, αVβ5. Data were fitted using the nonlinear curve fitting routines in Prism^®^ (Graphpad Software Inc, GraphPad Prism 5.0, San Diego, CA, USA) to obtain Kd values.

## Data Availability

The data presented in this study are available on request from the corresponding author. The data are not publicly available due to confidentiality of NEXEL’s data storage and distribution policy.

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
