# Peer review of "Truncated Milk Fat Globule-EGF-like Factor 8 Ameliorates Liver Fibrosis via Inhibition of Integrin-TGFβ Receptor Interaction"

_biomedicines, 2021, doi:10.3390/biomedicines9111529_

Round 1
Reviewer 1 Report
1. What are the correlation between TbR-I-integrin interaction and Smad phosphorylation and other signaling? Does NR-011 change cell surface receptor level (or localization) by antagonizing integrin function? or NR-011 inhibits TGF-beta bind to TBR-I by blocking integrin? In this study, there is a huge gap between anti-fibrosis and TGF-beta inhibition. Authors need to verify it.
2. Smad3 plays crucial role in ECM production and remodeling. Authors need to include Smad3 phosphorylation, not only Smad2.
3. Authors did not provide the detailed mechanism and biological functions of C2 domain of MGF-E8 in against TGF-beta signaling, and need to provide the rationale to remove C2 domain.
Author Response
We thank the reviewer for their thoughtful comments. We have done our best to address all concerns raised and provide a point-by-point response below, and the revised sentences were highlighted by blue color in the revised manuscript. We believe that the reviewer’s comments have significantly strengthened the revised manuscript.
Point 1: What are the correlation between TbR-I-integrin interaction and Smad
phosphorylation and other signaling? Does NR-011 change cell surface receptor level (or localization) by antagonizing integrin function? or NR-011 inhibits TGF-beta bind to TBR-I by blocking integrin? In this study, there is a huge gap between anti-fibrosis and TGF-beta inhibition. Authors need to verify it.
Response 1: Several integrins can interfere with both Smad-dependent and Smadindependent TGF-beta signaling in different ways, including the regulation of the expression of TGF-beta signaling pathway components, the physical association of integrins with TGFbeta receptors and the modulation of downstream effectors. For example, TGF-beta induced the association of integrin avb3 with TGF-βRII in both breast cancer cells and lung fibroblasts, initiating cooperative signaling to c-Src and MAPKs (J Biol Chem 279: 37726– 37733, Breast Cancer Res 8: R42). Similarly, TGF-βRII associates with avb5 in sclerodermal fibroblasts, and integrin signaling through FAK is necessary for TGF-beta-induced myofibroblastic differentiation (Am J Pathol 168: 499–510).
In this study, we found that the physical association of integrins with TGF-βRI in human hepatic stellate cells after TGF-β1 treatment (Figure 4D) and this association was further confirmed by immunoprecipitation assay (this data is now integrated in revised figure 4B). Furthermore, the down-stream molecule of TGF β signaling, Smad2, was phosphorylated with the increased association of integrins and TGF-βRI (Figure 4B). These physical association between integrins and TGF-βRI, and Smad2 phosphorylation were interfered by NP-011 treatment in human HSCs (Figure 4B ~ D). Thus, NP-011 regulates TGF β signaling
by interfering the physical association of integrins and TGF-βRI and modulates downstream effector of TGF-β signaling. The immunoprecipitation data was added to Figure 4B and the text was also modified accordingly.
In result section 3.4: [Proximity Ligation Assay (PLA) revealed the direct physical
associations between TGF-βRI and integrin β3 and β5, and the interactions became stronger upon TGF-β1 treatment in HSCs (Figure 4D,E). However, NP-011 treatment in TGF-β1- treated HSCs significantly loosen the associations between TGF- βRI and integrin β3 and β5 (Figure 4D,E). These physical associations were further confirmed by immunoprecipitation assay, and these patterns were identical to the Smad2 phosphorylation pattern by TGF and/or
NP-011 treatment. (Figure 4B). (Line 391-397)].
Point 2: Smad3 plays crucial role in ECM production and remodeling. Authors need to include Smad3 phosphorylation, not only Smad2.
Response 2: Thank you very much for this comment, and we agree with the Smad3 also plays a crucial role in ECM production and remodeling. However, in hepatic fibrosis, it was reported that the Smad2 and Smad3 play distinct roles in mediating liver fibrogenesis in human hepatic stellate cells (HSCs), and Smad2 protects TGF-b1/Smad3 mediated collagen synthesis (Mol Cell Biochem. 400(1-2):17-28). In the study, because the knockdown (K/D) or overexpression (O/E) of Smad3 showed opposite effects to the K/D or O/E of Smad2, this result implies that Smad2 could be a potential therapeutic target for the treatment of hepatic
fibrosis, thus, modulating Smad2 phosphorylation by NP-011 treatment in our manuscript would be sufficient to support the effect of NP-011 in treating hepatic fibrosis.
Point 3: Authors did not provide the detailed mechanism and biological functions of C2 domain of MGF-E8 in against TGF-beta signaling and need to provide the rationale to remove C2 domain.
Response 3: It is known that C2 domain of MFG-E8 could be a potential regulator of angiogenesis by promoting vascular remodeling (Genes Dev, 12, 21-33), and the C2 domain enables the MFG-E8 to bind to phosphatidylserine (PS) on apoptotic cells and integrin avb3/avb5 on phagocytic cells as a bridging molecule (Proc Natl Acad Sci U S A. 87:8417- 8421, Lancet. 351:1160-1164., Nature. 417:182-187.). Thus, there is no evidence regarding the known function of C2 domain in TGF beta signaling. By removing C2 domain in MFG-E8 in this study, the NP-011 might be considered to have a weaker binding to PS for phagocytosis, but it is thought that the binding affinity of integrin through the RGD motif was strengthened as a compensatory mechanism. In fact, the binding
affinity of NP-011 to integrin avb5 was significantly increased (> 10-fold) compared to the binding of MFG-E8 (Figure 4F and revised supplementary figure 2). Further, it was reported that the blockade of integrin avb5 reduces the TGFβ1 activation by 66 %, twice as much as a blockade of avb3 or b1 integrins (Nat Med 19:1617–1624). Therefore, rather than the C2 domain playing any role in the TGF-beta signaling, the removal of the C2 domain might compensate for stronger binding to the integrin avb5, thereby inhibiting the TGF-beta
signaling more effectively. We revised result and discussion session accordingly.
In result session 3.4: [Notably, the NP-011 showed about 12-fold stronger binding to integrin β5 than the binding of MFG-E8 (Kd of 25.4 nM, Supplementary figure 2). (Line 399 ~ 401)]. In discussion session: [The C2 domain of MFG-E8 facilitates it to bind to phosphatidylserine (PS) on apoptotic cells and integrin αvβ3/αvβ5 on phagocytic cells as a bridging molecule
[11,44-45]. By truncation of C2 domain in MFG-E8, the NP-011 might be considered to have a weaker binding to PS, but it might be strengthened the binding affinity of integrins through the RGD motif as a compensatory mechanism. In fact, the binding affinity of NP-011 to integrin αvβ5 was significantly increased (> 10-fold), compared to the binding of MFG-E8
(Figure 4F, S2). It is reported that the inhibition of integrin αvβ5 reduced the activation of TGF-β signaling by 66 %, twice as much as a blocking αvβ3 or αvβ1 integrins [46]. Therefore, the removal of the C2 domain in MFG-E8 protein might compensate for stronger binding to the integrin αvβ5, thereby inhibiting the TGF-beta signaling more effectively. (Line 441 ~ 450)]. Furthermore, the NP-011 has more benefits than the MFG-E8 as described in the discussion session. 1) The MFG-E8 is a glycoprotein that has glycosylation in the C2 domain (Biochem
Biophys Res Commun. 254:522–528, Biochim Biophys Acta. 6(1200):227–234).
Glycosylation in the protein therapy potentially induces immune response in the body (Nat Chem Biol. 2013 9: 776–784). Since the NP-011 has no glycosylation by removal of C2 domain of MFG-E8, the NP-011 has no concern about the potential immunogenicity due to the glycosylation in the body. 2) MFG-E8 contains medin site in the C2 domain known to cause Alzheimer's, type 2 diabetes, and aging, thus, the NP-011 also has no medin site in its sequences.
Reviewer 2 Report
This paper demonstrates the anti-fibrotic effect of a truncated form of MFG-E8 protein in a TAA-induced liver fibrosis model in mouse and an in vitro model consisting of APAP-induced human hepatocyte and HSC cells spheroids. The reduced fibrosis is supported by Sirius red staining, lower aSMA expression and completed in a transcriptomic analysis. The authors describe that the mechanism is driven by suppressing TGFb signaling that, among other effects, will reduce hepatic stellate cell (HSC) proliferation in a way av-class integrin-dependent. It is shown that TGFb-induced interactions between TGFb receptor I and integrins b3/5 are loosen by NP-011 presence. By proximity Ligation Assays the authors demonstrate that NP011 can bind avb3 and avb5 integrins. The results have a high interest and highlight the possibility of using this protein to treat fibrotic diseases. However, the data shown here are not sufficiently convincing to demonstrate an enhanced effect of NP-011 compared to the complete molecule. Moreover, with the proposed mechanism of action is hard to understand the enhanced effect of NP-011 compared to MFG-E8. Other important aspect is that the truncation could eliminate important adverse effects of the complete molecule, but there are no data showing lack of adverse actions in NP-011.
- The Coomassie staining in Fig 1A shows an intense band of about 15 kDa that is contaminating batch 1. Then, this staining apparently does not show a >95% purity as indicated in the text. Do the authors know the identity of this contaminant molecule? Can establish that is not interfering with their analysis?
- Comparison between MFG-E8 and NP-011 in the in vivo liver fibrosis model (Fig 1C-E) only shows light but significant difference in the fibrotic area but not in Col I expression. Then, is a bit excessive to conclude that “deletion of C2 domain gives more powerful beneficial effects on curing liver fibrosis than the original protein”.
- In the rest of analysis lacks comparison between the two proteins, with the exception of TGFb signaling in fig 4A. Here NP-011 has a clear effect during 1 hour but after 2 hours this effect is reduced and was less effective than MFG-E8. How do you explain this result? Possibly this implies that in vivo the two molecules act similarly on TGFb signaling.
- It is demonstrated that NP011 can bind avb3 and avb5 integrins and that Cilengitide reverts the suppressive effect of NP-011 on HSC proliferation, thus suggesting that its RGD motif is implicated. In principle the RGD motif is also present in MFG-E8. Therefore, is unclear the more intense suppressive effect of NP-011 on TGFb signaling compared to MFG-E8. In order to understand if the supposed stronger action of NP-011 is due to its activity to bind avb3 and avb5 integrins, the authors should analyze and compare the binding affinity of MFG-E8 to the same integrins and show in results. In addition, they should identify the region of the molecule that is involved in this action.
- Do any of the parameters studied in the part of safety profiles gives negative values in animals treated with MFG-E8? If not, this should be mentioned in the text. If there are differences with NP-100, this result should be shown.
- It was published that MFG-E8 binds collagen facilitating collagen uptake by macrophages and then ameliorating pulmonary fibrosis. Does NP-011 bind collagen and could have a contribution in the anti-fibrotic effect in this model? The authors should add this information in the paper.
Minor
- Please check the first sentence in the abstract
- The paper would improve if the authors add a cartoon with the parts/domains of the molecule and those that are eliminated in the truncated form.
Author Response
We thank the reviewer for their thoughtful comments. We have done our best to address all concerns raised and provide a point-by-point response below, and the revised sentences were highlighted by blue color in the revised manuscript. We believe that the reviewer’s comments have significantly strengthened the revised manuscript. Please see the attachment.

Round 2
Reviewer 2 Report
The authors answered all the points and concerns of this reviewer about the manuscript.